# BurstAttention: An Efficient Distributed Attention Framework for Extremely Long Sequences

## Abstract

Effective attention modules have played a crucial role in the success of Transformer-based large language models (LLMs), but the quadratic time and memory complexities of these attention modules also pose a challenge when processing long sequences. One potential solution for the long sequence problem is to utilize distributed clusters to parallelize the computation of attention modules across multiple devices (e.g., GPUs). However, adopting a distributed approach inevitably introduces extra memory overheads to store local attention results and incurs additional communication costs to aggregate local results into global ones. In this paper, we propose a distributed attention framework named "BurstAttention" to optimize memory access and communication operations at both the global cluster and local device levels. In our experiments, we compare BurstAttention with other competitive distributed attention solutions for long sequence processing. The experimental results under different length settings demonstrate that BurstAttention offers significant advantages for processing long sequences compared with these competitive baselines, reducing 40% communication overheads and achieving $2\times$ speedup during training 128K sequence length on $8\times$A100.

## 1 Introduction

Transformers (Vaswani et al., 2017) have emerged as the dominant architectures for large language models (LLMs) (Brown et al., 2020; Chowdhery et al., 2022) due to their remarkable capacities to understand complex text and generate controllable responses. Empirically, the power of Transformers lies largely in their multi-head attention modules, which enable Transformers to capture rich semantic information from textual contexts effectively. For every plus, there is a minus. Despite the success of Transformers' attention modules, these modules exhibit quadratic time and memory complexity concerning sequence length, posing challenges in terms of both computing time and memory overheads as sequence length increases.

Various efforts have been devoted to making attention modules more efficient and enabling LLMs to process longer sequences. One direction is taking full advantage of a single device's compute and storage units (e.g., a GPU) to process long sequences, such as FlashAttention (Dao et al., 2022). FlashAttention can significantly accelerate the computation of attention modules by using more efficient static random access memory (SRAM) instead of high-bandwidth memory (HBM) in devices to store intermediate attention states. Another direction is using distributed clusters containing multiple devices (e.g., multiple GPUs) to process long sequences, such as RingAttention (Li et al., 2021). RingAttention divides long sequences into multiple subsequences and processes subsequences separately on different devices. Besides these efforts, some lossy methods, such as sparse attention methods (Zaheer et al., 2020; Ding et al., 2023), are also widely explored to reduce the computing time and memory requirements of attention modules within a tolerable performance penalty.

All the above improvements orienting to improve attention modules have achieved promising results, and an intuitive problem is raised — whether we can combine these improvements to achieve a more efficient attention solution. This paper introduces an efficient distributed attention framework to handle extremely long sequences named "BurstAttention". BurstAttention can take full advantage of the power of both distributed clusters and single devices while being compatible with lossy sparse

attention methods. Specifically, given an extremely long sequence, BurstAttention first divides the sequence into partitions according to the number of devices in distributed clusters, and each partition is assigned to one of these devices. Then, each device projects the partitioned sequence into query, value, and key embedding partitions. The query partitions are pinned, and all key-value partitions are passed through all devices to compute their local attention scores with each pinned query partition. Based on the local attention scores, a global attention operation is adopted to aggregate the local results into the final global results.

By fine-grained scheduling the computation and communication operations of devices during computing attention modules, as well as introducing online softmax operations (Milakov & Gimelshein, 2018), BurstAttention proposes global attention optimization (GAO) and local attention optimization (LAO) strategies, which can fully optimize the input-output (I/O) and communication procedures in distributed clusters. These two strategies offer substantial benefits for computing local attention scores in each device and aggregating local results into global ones in the whole cluster, including improved memory consumption, reduced communication overhead, and enhanced cache utilization. Since BurstAttention splits sequences into multiple partitions for processing, this design naturally makes it adaptable to any optimization strategies at the local attention level, especially the above-mentioned sparse attention methods (Zaheer et al., 2020; Ding et al., 2023). Also, owing to just splitting sequences, BurstAttention is orthogonal to other distributed methods and can be easily integrated with these for training and inference Transformer-based LLMs, such as data parallelism (Valiant, 1990), tensor parallelism (Narayanan et al., 2021), pipeline parallelism (Huang et al., 2019), and zero redundancy optimizer (Rajbhandari et al., 2020; Ren et al., 2021).

We evaluate BurstAttention and current competitive distributed attention solutions (Dao et al., 2022; Li et al., 2021) under various sequence length settings. The experimental results show that BurstAttention is a memory-efficient solution for attention modules to process long sequences and achieve good data throughputs. Moreover, since BurstAttention greatly optimizes the communication operations in the computation process of attention modules, BurstAttention makes it more difficult for device communication to become a bottleneck as the devices in distributed clusters increase, and thus can take better advantage of distributed clusters than other attention solutions.

## 2 RELATED WORK

Transformer-based LLMs such as GPT (Brown et al., 2020; Ouyang et al., 2022), LLaMA (Touvron et al., 2023a;b), and PaLM (Chowdhery et al., 2022; Anil et al., 2023) have achieved great success in recent years (Han et al., 2021; Bommasani et al., 2021; Zhao et al., 2023). Despite the success of these LLMs, they still face efficiency challenges: one is that as these models continue to grow in size, the computational and memory costs associated with training and inference have become bottlenecks. Another is that the quadratic attention computational complexity of the Transformer architecture makes these LLMs difficult to handle long sequences. Up to now, various parallelism strategies (Valiant, 1990; Huang et al., 2019; Rajbhandari et al., 2020; Narayanan et al., 2021) and memory optimization strategies (Ren et al., 2021; Chen et al., 2016; Korthikanti et al., 2023), which have significantly improved the training and inference efficiency of LLMs, have well solved the computational bottleneck caused by the model size growth, but it is still challenging to solve the efficiency issue caused by the sequence growth.

To enable LLMs to process longer sequences more efficiently, several attention solutions have been proposed. Korthikanti et al. (2023) adopt selective activation recomputation to avoid storing attention softmax logits during the forward pass, and then recompute these logits during the backward pass to build a computation graph for backpropagation, significantly reducing memory overheads of attention modules to process long sequences. Rabe & Staats (2021) formalize the computation of attention modules at the block level and make each thread block in devices handle the attention computation of a sub-sequence, further reducing temporary memory consumptions and achieving a logarithmic memory complexity relative to the sequence length. Based on these works, Dao et al. (2022) introduce FlashAttention, a CUDA implementation of attention modules that leverages the fast I/O capabilities of the SRAM in devices for further speedup. FlashAttention optimizes the attention algorithm by introducing I/O complexity analysis and minimizing the I/O costs on the HBM in devices, offering a new perspective on attention optimization.

While the above solutions focus on optimizing the long-sequence attention problem using a single device, they still struggle to handle extremely long sequences due to the limitations of a single device's performance. Some recent efforts have therefore aimed to address this long-sequence challenge using distributed clusters, i.e., using multiple devices. The most straightforward method is to use general parallelism strategies, such as data parallelism (Valiant, 1990), tensor parallelism (Narayanan et al., 2021), pipeline parallelism (Huang et al., 2019), and zero redundancy optimizer (Rajbhandari et al., 2020; Ren et al., 2021). In order to better use distributed clusters for attention modules to process long sequences, Li et al. (2021) propose sequence parallelism method RingAttention, which splits the computation and memory overheads of attention modules across multiple devices following the sequence dimension.

Various sparse attention methods, including low-rank methods (Winata et al., 2020; Wang et al., 2020), kernel-based methods (Katharopoulos et al., 2020; Choromanski et al., 2020; Qin et al., 2022) and downsampling methods (Lee et al., 2019; Jaegle et al., 2021) are also widely explored. These methods reduce the time and memory requirements of attention modules by computing a limited selection of similarity scores from a sequence rather than all possible pairs, resulting in sparse attention softmax logits rather than dense ones. Recently, Ding et al. (2023) have explored implementing sparse attention methods based on distributed clusters and achieved promising results. Note that these sparse attention methods inevitably lead to significant performance degradation, along with reducing the time and memory requirements. In the actual processing of long sequences, the use of these lossy methods needs to be cautious.

Existing attention solutions to process long sequences mainly focus on one specific optimization aspect. This paper provides a holistic perspective that encompasses all the above-mentioned aspects and offers an efficient distributed attention framework to process extremely long sequences.

## 3 METHODOLOGY

### 3.1 PRELIMINARY

As the key module in Transformers (Vaswani et al., 2017), an attention module can be formalized as

$$\mathbf{S} = \frac{\mathbf{Q}\mathbf{K}^T}{\sqrt{d}}, \quad \mathbf{P} = \text{softmax}(\mathbf{S}), \quad \mathbf{O} = \mathbf{P}\mathbf{V}, \tag{1}$$

where $\mathbf{Q} \in \mathbb{R}^{N \times d}$ indicates the embeddings of the query sequence, $N$ is the length of the query sequence, and $d$ is the embedding dimension. $\mathbf{K} \in \mathbb{R}^{N \times d}$ and $\mathbf{V} \in \mathbb{R}^{N \times d}$ indicate the embeddings of the key sequence and the value sequence, respectively. $\mathbf{S} \in \mathbb{R}^{N \times N}$ is the attention score, $\mathbf{P} \in \mathbb{R}^{N \times N}$ is the attention probability. $\mathbf{O} \in \mathbb{R}^{N \times d}$ is the final attention result, which is the average of the value sequence embeddings weighted by the similarities between the query sequence and the key sequence. In this paper, we mainly use self-attention modules to illustrate BurstAttention, but BurstAttention can be easily extended to cross-attention modules. For more details of various attention modules in the Transformer architecture, we recommend referring to the original paper of Transformers (Vaswani et al., 2017), and we will not go into details here.

### 3.2 THE WHOLE FRAMEWORK OF BURSTATTENTION

We build the whole framework of BurstAttention based on sequence parallelism (Li et al., 2021), where $\mathbf{Q}$, $\mathbf{K}$ and $\mathbf{V}$ are divided into multiple partitions along the sequence dimension according to the number of devices (e.g., GPUs) in a distributed cluster. Each device in the cluster will be assigned a query partition, a key partition, and a value partition. Formally, given the device number $G$, the $i$-th device will be assigned $\mathbf{Q}_i, \mathbf{K}_i, \mathbf{V}_i \in \mathbb{R}^{\frac{N}{G} \times d}$. As shown in Figure 1, at each step, the $i$-th device receives a key partition $\mathbf{K}_j$ and a value partition $\mathbf{V}_j$ from its previous neighbor and performs local attention operations. After that, the $i$-th device sends its received key and value partitions $\mathbf{K}_j$ and $\mathbf{V}_j$ to its next neighbor for the use of the next step, which forms a ring-style communication process. This ring-style communication process continues until all $\mathbf{K}$ and $\mathbf{V}$ partitions have made a full circle around the ring, completing local attention operations on all devices. The local attention operations can be formalized as

$$\mathbf{S}_{i,j} = \frac{\mathbf{Q}_i\mathbf{K}_j^T}{\sqrt{d}}, \quad \mathbf{P}_{i,j} = \text{softmax}(\mathbf{S}_{i,j}), \quad \mathbf{O}_{i,j} = \mathbf{P}_{i,j}\mathbf{V}_j, \tag{2}$$

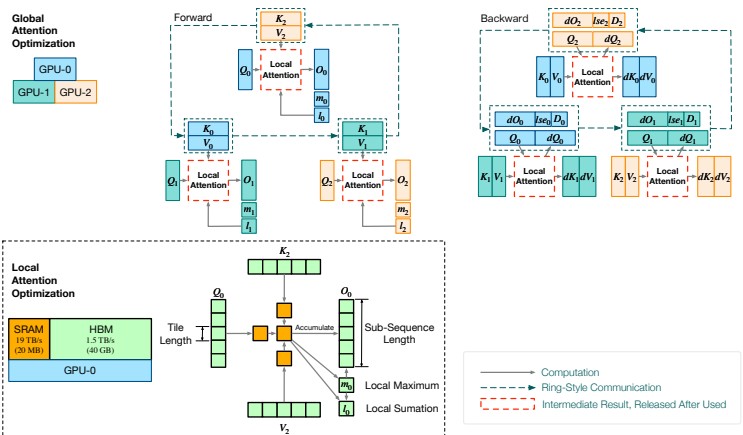

Figure 1: In this figure, we undertake a two-step partitioning of the sequence input: first, dividing it across multiple devices (inter-device), and then further splitting it within each single device (intra-device). First, We partition the query, key, and value across multiple devices and pass the sliced sequence through each device in a ring-like communication, allowing each device to process only a local attention at a time. This avoids the burden on memory caused by processing extremely long sequence at once. We then aggregate local attention results into global attention results. By transmitting $\mathbf{K}, \mathbf{V}$ simultaneously, we avoid storing intermediate result $\mathbf{QK}^T$, which has quadratic memory complexity, and instead recompute it during the backward pass, which we call global attention optimization (GAO). In local attention, we further partition the sub-sequence into smaller tiles, aiming to perform block-wise computations within the device. This allows us to take advantage of the high bandwidth of SRAM while minimizing access to the lower bandwidth HBM, which we call local attention optimization (LAO).

where $\mathbf{O}_{i,j} \in \mathbb{R}^{\frac{N}{G} \times d}$ is the local attention results between the device-assigned query partition $\mathbf{Q}_i$ and the device-received partitions $\mathbf{K}_j$ and $\mathbf{V}_j$. $\mathbf{S}_{i,j} \in \mathbb{R}^{\frac{N}{G} \times \frac{N}{G}}$ is the local attention score, and $\mathbf{P}_{i,j} \in \mathbb{R}^{\frac{N}{G} \times \frac{N}{G}}$ is the local attention probability. Obviously, Eq. (1) and Eq. (2) are not equivalent, we thus introduce global attention operations to aggregate all local attention results $\{\mathbf{O}_{i,j}\}_{i=1,j=1}^{\frac{N}{G},\frac{N}{G}}$ into the final partitioned attention results $\mathbf{O}_i \in \mathbb{R}^{\frac{N}{G} \times d}$, and $\{\mathbf{O}_i\}_{i=1}^{\frac{N}{G}}$ is the final global attention results. To make both the global and local attention operations more efficient, we introduce Global Attention Optimization (GAO) and Local Attention Optimization (LAO), respectively. Next, we will introduce how to perform these attention optimization strategies in detail.

### 3.3 GLOBAL ATTENTION OPTIMIZATION (GAO)

For global attention operations, the main idea is to aggregate $\mathbf{O}_{i,j}$ into $\mathbf{O}_i$. For some conventional methods such as RingAttention (Li et al., 2021), for the $i$-th query partition, they store the intermediate results $\mathbf{S}_{i,j}$ and $\mathbf{P}_{i,j}$ for every $j$ throughout the ring-style communication process. This introduces a non-negligible memory overhead. To get rid of this memory overhead, we introduce GAO.

As shown in Figure 1, GAO consists of two main steps. First, similar to RingAttention, devices are organized in a ring for communication. Each round, $\mathbf{K}, \mathbf{V}$ partitions are shifted along the ring to the next adjacent device. Second, after each round of $\mathbf{K}, \mathbf{V}$ transmission, each device $i$ performs a local attention operation using the partitions $\mathbf{Q}_i$ and its received partition $\mathbf{K}_j$, and $\mathbf{V}_j$, as described in Eq. (2). The local attention result $\mathbf{O}_{i,j}$ are then dynamically accumulated into global attention result $\mathbf{O}_i$ by employing online softmax (Milakov & Gimelshein, 2018), which eliminates the need to store intermediate results $\mathbf{S}_{i,j}$ and $\mathbf{P}_{i,j}$.

As depicted in Algorithm 1, in the forward pass, we dynamically maintain the row-wise maximum value $m_i$ of $\mathbf{S}_{\mathbf{i},\mathbf{j}}$ as in Line 11 and the row-wise sum $l$ of $\mathbf{P}_{\mathbf{i},\mathbf{j}}$ as in Line 12 to avoid storing $\mathbf{S}$ and $\mathbf{P}$, and use $m_i$ and $l_i$ for scaling during the aggregation of $\mathbf{O}_i$ as in Line 13. Note that, the functions rowmax$(\cdot)$ and rowsum$(\cdot)$ can be formalized as

$$[\text{rowmax}(\mathbf{W})]_i = \max_j \{[\mathbf{W}]_{i,j}\}, \quad [\text{rowsum}(\mathbf{W})]_i = \sum_j [\mathbf{W}]_{i,j}, \tag{3}$$

---

**Algorithm 1:** The forward pass of GAO

---

**Data:** Matrices $\mathbf{Q}_i, \mathbf{K}_i, \mathbf{V}_i \in \mathbb{R}^{\frac{N}{G} \times d}$ on the $i$-th device

1  Initialize $\mathbf{O}_i = (0)_{\frac{N}{G} \times d} \in \mathbb{R}^{\frac{N}{G} \times d}, l_i = (0)_{\frac{N}{G}} \in \mathbb{R}^{\frac{N}{G}}, m_i = (-\infty)_{\frac{N}{G}} \in \mathbb{R}^{\frac{N}{G}}$;

2  Put $\mathbf{K}_i, \mathbf{V}_i$ into communication ring;

3  **for** $j = 1$ *to* $G$ **do**

4      Conduct one step of ring communication;

5      Get $\mathbf{K}_j, \mathbf{V}_j$ from communication ring;

    /* The forward pass of local attention operations (w/o LAO).               */

6      $\mathbf{S}_{i,j} = \mathbf{Q}_i \mathbf{K}_j^T$;

7      $m_{i,j} = \text{rowmax}(\mathbf{S}_{i,j})$;

8      $\mathbf{P}_{i,j} = \exp(\mathbf{S}_{i,j} - m_{i,j})$;

9      $l_{i,j} = \text{rowsum}(\mathbf{P}_{i,j})$;

10     $\mathbf{O}_{i,j} = \mathbf{P}_{i,j} \mathbf{V}_j$;

    /* The end of the forward pass of local attention operations.         */

11     $m_{\text{new}} \leftarrow \max\{m_i, m_{i,j}\}$;

12     $l_i = e^{m_i - m_{\text{new}}} l_i + e^{m_{i,j} - m_{\text{new}}} l_{i,j}$;

13     $\mathbf{O}_i = e^{m_i - m_{\text{new}}} \mathbf{O}_i + e^{m_{i,j} - m_{\text{new}}} \mathbf{O}_{i,j}$;

14     $m_i = m_{\text{new}}$;

15     Put $\mathbf{K}_j, \mathbf{V}_j$ into communication ring;

16 $\mathbf{O}_i = \text{diag}(l_i)^{-1} \mathbf{O}_i$;

17 $lse_i = m_i + \log l_i$;

18 Return $\mathbf{O}_i, lse_i$;

---

**Algorithm 2:** The backward pass of GAO

---

**Data:** Matrices $\mathbf{Q}_i, \mathbf{K}_i, \mathbf{V}_i, \mathbf{O}_i, \mathbf{dO}_i \in \mathbb{R}^{\frac{N}{G} \times d}, lse_i \in \mathbb{R}^N$ on the $i$-th device

1  Initialize $\mathbf{dQ}_i, \mathbf{dK}_i, \mathbf{dV}_i = (0)_{\frac{N}{G} \times d} \in \mathbb{R}^{\frac{N}{G} \times d}$;

2  $D_i = \text{rowsum}(\mathbf{dO}_i \circ \mathbf{O}_i)$ (pointwise multiply);

3  Put $\mathbf{Q}_i, \mathbf{dQ}_i, \mathbf{dO}_i, D_i, lse_i$ into communication ring;

4  **for** $j = 1$ *to* $G$ **do**

5      Conduct one step of ring communication;

6      Get $\mathbf{Q}_j, \mathbf{dQ}_j, \mathbf{dO}_j, D_j, lse_j$ from communication ring;

    /* The backward pass of local attention operations (w/o LAO).           */

7      $\mathbf{S}_{j,i} = \mathbf{Q}_j \mathbf{K}_i^T$;

8      $\mathbf{P}_{j,i} = \exp(\mathbf{S}_{j,i} - lse_j)$;

9      $\mathbf{dV}_i = \mathbf{dV}_i + \mathbf{P}_{j,i}^T \mathbf{dO}_j$;

10     $\mathbf{dP}_{j,i} = \mathbf{dO}_j \mathbf{V}_i^T$;

11     $\mathbf{dS}_{j,i} = \mathbf{P}_{j,i} \circ (\mathbf{dP}_{j,i} - D_j)$;

12     $\mathbf{dK}_i = \mathbf{dK}_i + \mathbf{dS}_{j,i}^T \mathbf{Q}_j$;

13     $\mathbf{dQ}_j = \mathbf{dQ}_j + \mathbf{dS}_{j,i} \mathbf{K}_i$;

    /* The end of the backward pass of local attention operations.        */

14     Put $\mathbf{Q}_j, \mathbf{dQ}_j, \mathbf{dO}_j, D_j, lse_j$ into communication ring;

15 Return $\mathbf{dQ}_G, \mathbf{dK}_G, \mathbf{dV}_G$;

---

where $[\cdot]_i$ is the $i$-th element of the vector, $[\cdot]_{i,j}$ is the element in the $i$-th row and $j$-th column of the matrix. Considering the requirements of the backward pass, we also store $lse_i$ besides the global attention results $\mathbf{O}_i$ after the forward pass, which can make the subsequent backward pass more efficient. During the backward pass, as depicted in Algorithm 2, we employ the same strategy for the forward pass to obtain gradients based only on recomputed $\mathbf{S}, \mathbf{P}$ and output information.

### 3.4 LOCAL ATTENTION OPTIMIZATION (LAO)

Given $\mathbf{Q}_i, \mathbf{K}_j$, and $\mathbf{V}_j$, the local attention operations that involve these partitions are performed only on a single device (e.g., a GPU). When computing $\mathbf{O}_{i,j}$ in Eq. (2), $\mathbf{S}_{i,j}$ and $\mathbf{P}_{i,j}$ are computed and stored on the HBM of the device. To avoid frequent I/O operations of $\mathbf{S}_{i,j}$ and $\mathbf{P}_{i,j}$ on the HBM, the local attention operations of BurstAttention, inspired from FlashAttention (Dao et al., 2022), further divide $\mathbf{Q}_i, \mathbf{K}_j$, and $\mathbf{V}_j$ into tiles along the sequence dimension, with each tile $\frac{M}{4d}$ sequence length, where $M$ represents the SRAM size of the device, $d$ represents the attention head dimension.

As shown in Figure 1, during computing $\mathbf{O}_{i,j}$, each thread block reads the tiles of $\mathbf{Q}_i, \mathbf{K}_j, \mathbf{V}_j$ from the HBM to SRAM, the tiles of $\mathbf{S}_{i,j}$ and $\mathbf{P}_{i,j}$ are computed and then written on the SRAM instead of the HBM, $\mathbf{O}_{i,j}$ are dynamically accumulated based on online softmax operations and written back to the HBM. Since the SRAM has a much higher I/O bandwidth than the HBM, the above optimization can make local attention operations more efficient. Although the memory of the SRAM is tiny, further

| Method | FlashAttention/LAO | Memory | | Communication | |
|---|---|---|---|---|---|
| | | Parameter | Activation | Forward | Backward |
| RingAttention | w/o | $4HZd$ | $4\frac{BZNd}{G} + \frac{BZN^2}{G} + \frac{BNH}{G}$ | $2BZNd$ | $6BZNd$ |
| RingAttention$^\dagger$ | – | – | – | | |
| Tensor Parallelism | w/o | $4\frac{HZd}{G}$ | $4\frac{BZNd}{G} + \frac{BZN^2}{G} + BNH$ | $4BZNd$ | $4BZNd$ |
| Tensor Parallelism | w/ | | $4\frac{BZNd}{G} + \frac{BZN^2}{(M/4d)G} + BNH$ | | |
| BurstAttention | w/o | $4HZd$ | $4\frac{BZNd}{G} + \frac{BZN^2}{G^2} + \frac{BNH}{G}$ | $2BZNd$ | $3BZNd$ |
| BurstAttention | w/ | | $4\frac{BZNd}{G} + \frac{BZN^2}{(M/4d)^2G^2} + \frac{BNH}{G}$ | | |

Table 1: The memory and communication overheads of various distributed attention solutions. $G$ is the device number of the whole distributed cluster, $B$ denotes the batch size, $N$ represents the sequence length, $Z$ signifies the number of attention heads, $d$ corresponds to the hidden dimension per head, $H$ represents the model dimension of Transformers, and $M$ represents the device SRAM size. $^\dagger$ means from an implementation perspective, RingAttention's separating $\mathbf{K}$ and $\mathbf{V}$ into two independent rounds of communication cannot be combined with FlashAttention to improve efficiency.

dividing $\mathbf{Q}_i$, $\mathbf{K}_j$, and $\mathbf{V}_j$ into many fine-grained tiles ensure the intermediate results $\mathbf{S}_{i,j}$ and $\mathbf{P}_{i,j}$ can be entirely stored into the SRAM.

Intuitively, when BurstAttention is running on a single device rather than a distributed cluster, there is no need to use GAO at this time, and LAO will play the same role as FlashAttention. In other words, FlashAttention can be viewed as a specialization of BurstAttention on a single device.

### 3.5 INTEGRATING BURSTATTENTION WITH SPARSE ATTENTION METHODS

As mentioned before, the sequence parallelism mechanism makes BurstAttention easy to cooperate with sparse attention methods. During the computation process of BurstAttention, given $\mathbf{Q}_i$, $\mathbf{K}_j$, $\mathbf{V}_j$, if there is no need to compute the similarities between these partitions, then the local attention operations on these partitions can be skipped directly. If just some tokens in $\mathbf{Q}_i$, $\mathbf{K}_j$ and $\mathbf{V}_j$ are required to compute their similarities for final attention results, we can similarly skip unnecessary operations in local attention operations.

## 4 ANALYSIS

In this section, we will analyze the memory, I/O, and communication overheads of BurstAttention as compared to existing competitive distributed attention solutions. As data parallelism and pipeline parallelism are often used as the most basic distributed strategies and cannot reduce the cost of long sequence processing, we focus here on comparing BurstAttention, tensor parallelism (Narayanan et al., 2021), and the typical sequence parallelism method RingAttention (Li et al., 2021).

### 4.1 MEMORY AND I/O OVERHEADS

In terms of memory complexity, when we split the input along the sequence dimension across devices for global operations and further split them in each device for local operations, the memory overheads caused by $\mathbf{Q}\mathbf{K}^T$ will be reduced to $\frac{1}{(M/d)^2 G^2}$ of the original ones. Table 1 shows the memory overheads of various distributed attention solutions. The table shows that BurstAttention has lower activation memory while tensor parallelism has lower parameter memory. This means that the longer the sequence, the more pronounced the advantage of BurstAttention. Moreover, by combining BurstAttention with some parallelism strategies like zero redundancy optimizer (Rajbhandari et al., 2020; Ren et al., 2021) to partition parameters, BurstAttention can easily obtain the same parameter memory overheads as tensor parallelism. In terms of I/O overheads, RingAttention requires $\Theta(\frac{BZN^2}{G} + BZNd)$ memory accesses on every single device of the whole cluster; tensor parallelism and BurstAttention only requires $\Theta(\frac{BZN^2}{(M/d^2)G})$ memory accesses. This indicates that BurstAttention can significantly reduce I/O time costs compared to other distributed attention baselines.

### 4.2 COMMUNICATION OVERHEADS

In the forward pass, BurstAttention involves one round of ring-style peer-to-peer communications on the $\mathbf{K}, \mathbf{V} \in \mathbb{R}^{B \times Z \times \frac{N}{G} \times d}$, with a total cost of $\Theta(2BZNd)$. In the backward pass, BurstAttention

Table 2: The first token latency of the LLaMA-7b inference (s).

| Sequence Length | 4,096 | 8,192 | 16,384 | 32,768 | 65,536 | 131,072 | 262,144 |
|---|---|---|---|---|---|---|---|
| RingAttention | 0.42±0.01 | 0.87±0.01 | 2.00±0.01 | 5.13±0.05 | OOM | OOM | OOM |
| TP(Megatron V1) w/ Flash | 0.67±0.01 | 1.29±0.01 | 2.58±0.01 | 5.27±0.01 | 11.63±0.02 | 27.54±0.01 | 71.52±0.06 |
| TP(Megatron V3) w/ Flash | 0.73±0.02 | 1.36±0.01 | 2.68±0.01 | 5.67±0.01 | 12.25±0.01 | 28.73±0.03 | 75.52±0.05 |
| BurstAttention w/o LAO | 0.46±0.01 | 0.88±0.01 | 1.79±0.01 | 3.88±0.01 | 10.78±0.01 | OOM | OOM |
| BurstAttention | **0.44±0.01** | **0.84±0.01** | **1.68±0.01** | **3.27±0.01** | **6.49±0.01** | **16.01±0.01** | **49.32±0.11** |

Table 3: The first token latency of the LLaMA-13b inference (s).

| Sequence Length | 4,096 | 8,192 | 16,384 | 32,768 | 65,536 | 131,072 | 262,144 |
|---|---|---|---|---|---|---|---|
| RingAttention | 0.66±0.01 | 1.36±0.01 | 3.08±0.01 | 7.98±0.02 | OOM | OOM | OOM |
| TP(Megatron V1) w/ Flash | 1.05±0.01 | 2.01±0.01 | 4.03±0.01 | 8.41±0.01 | 18.56±0.02 | 44.39±0.04 | OOM |
| TP(Megatron V3) w/ Flash | 1.07±0.01 | 2.09±0.01 | 4.20±0.01 | 8.76±0.01 | 19.06±0.06 | 45.46±0.03 | 119.03±0.04 |
| BurstAttention w/o LAO | 0.72±0.01 | 1.39±0.01 | 2.77±0.05 | 5.99±0.01 | 16.95±0.01 | OOM | OOM |
| BurstAttention | **0.69±0.01** | **1.40±0.05** | **2.57±0.03** | **5.08±0.02** | **9.92±0.01** | **25.91±0.01** | **78.80±0.07** |

requires one round of ring-style communication on tensors $\mathbf{Q}, \mathbf{dQ}, \mathbf{dO} \in \mathbb{R}^{B \times \frac{N}{G} \times Z \times d}$ and $D, lse \in \mathbb{R}^{B \times \frac{N}{G} \times Z}$, with a total cost of $\Theta(3BZNd + 2\frac{BNZ}{G})$. Table 1 shows the communication overheads of various distributed attention solutions. The forward communication of RingAttention is the same as BurstAttention, which is $\Theta(2BZNd)$, but without GAO and LAO, RingAttention requires a total cost of $\Theta(6BZNd)$ in the backward pass, which is about twice that of BurstAttention. Therefore, BurstAttention has great advantage of communication overheads during training than RingAttention. The forward communication of tensor parallelism is $\Theta(4BZNd)$ and the total communication is $\Theta(8BZNd)$, thus BurstAttention also has higher communication efficiency during both inferring and training than tensor parallelism.

# 5 EXPERIMENTS

## 5.1 EXPERIMENTAL SETTINGS

We conduct our experiments on a distributed cluster of $8 \times$A100 GPUs interconnected by PCI-E. We use two LLMs in our experiments, LLaMA-2 with 7 billion parameters (7b) and LLaMA-2 with 13 billion parameters (13b) (Touvron et al., 2023b).

Our experiments consist of five methods: (1) **TP**, which refers to tensor parallelism (Narayanan et al., 2021), a commonly used distributed strategy in the stages of both training and inference. Note that here we futher classify TP into **TP(Megatron V1)** and **TP(Megatron V3)** based on the detail communication operations (Megatron V1 uses all-reduce while Megatron V3 uses the combination of all-gather and reduce-scatter). (2) **TP w/ FlashAttention**, which combines FlashAttention (Dao et al., 2022) with tensor parallelism as a strong baseline. **Note that this is a commonly used strategy in current LLM pre-training and inference.** (3) **RingAttention**, a typical sequence parallelism baseline. (4) **BurstAttention**, our distributed attention method includes both GAO and LAO strategies. (5) **BurstAttention w/o LAO**, where we remove the LAO strategy for ablation studies. (6) **BurstAttention+ZeRO** , where we futher optimize the memory overhead of BurstAttention by adopting the ZeRO(Rajbhandari et al., 2020) technique to shard model parameters across devices.

As we mentioned before, data parallelism and pipeline parallelism cannot effectively reduce the cost of long sequence processing, and we do not use them as baselines. In fact, we conduct some experiments to adapt data parallelism and pipeline parallelism for long-sequence attention, but unfortunately, these two parallelism methods cannot process extremely long sequences. **From our pilot experiments, directly adopting data parallelism or pipeline parallelism can only handle sequences shorter than 8192, much shorter than RingAttention and TP.**

## 5.2 INFERENCE LATENCY

In this section, we focus on the latency needed for generating the first token (i.e., the first token latency) in the inference process. We concentrate on the time of the first token generation because the long sequence attention computation mainly exists in the inference encoding process. Since the first

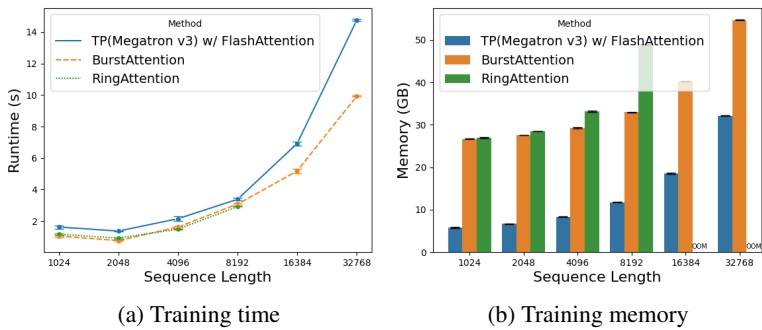

(a) Training time

(b) Training memory

Figure 2: The training time and memory of LLaMA-7b on 8×A100.

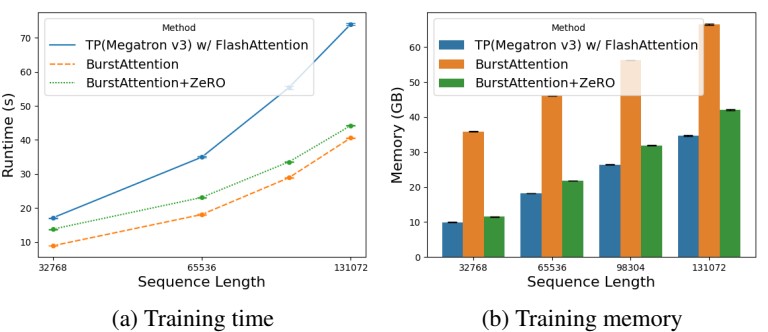

(a) Training time

(b) Training memory

Figure 3: The training time and memory of LLaMA-7b on 32×A100.

token latency is much higher than the latency of generating subsequent tokens, the first token latency thus becomes one of the most critical targets existing works seek to optimize.

In real-time AI services such as ChatGPT, the system's responsiveness significantly impacts the user experience, and these applications usually output results in a streaming manner to improve responsiveness. Since the first token latency is the longest, the first token latency directly influences the perceived responsiveness and efficiency of the model in these streaming scenarios.

As shown in Table 2 and Table 3, we can see that, compared with tensor parallelism, sequence parallelism methods are more suitable to infer long sequences. Compared with the RingAttention method, by using GAO, BurstAttention can support longer sequences. By further using LAO, BurstAttention can achieve more latency improvements and support much longer sequences. Note that, although TP(Megatron V3) is more memory efficient than TP(Megatron V1), the all-reduce operation used by TP(Megatron V1) is better optimized than the reduce-scatter and all-gather operations used by TP(Megatron V3). In the actual inference, TP(Megatron V1) is slightly faster than TP(Megatron V3). Since TP(Megatron V3) has a similar time to TP(Megatron V1) but better memory efficiency, we mainly compare our method with TP(Megatron V3) in subsequent experiments.

## 5.3 TRAINING PERFORMANCE

For training LLMs, a batch is required to have 2 to 4 million tokens, otherwise, the model performance may be degraded, i.e., the longer the sequence length is, the smaller the batch size is. Due to this, several GPUs may need to process one example together. For example, using 2048 GPUs to train 128-layer GPT-3, the sequence length is 4096, the batch size is 1024, data parallelism is 16, pipeline parallelism is 32, and tensor parallelism is 4. In this scenario, the optimal setup is to divide a batch into 64 micro-batches with a micro-batch size of 1. In this case, four GPUs under the same tensor parallelism group are inevitably required to process one piece of data together. In view of this, we fix the batch size to 1 for experimental convenience and vary the input sequence length from 1K to 32K.

As can be seen from Figure 2a, although tensor parallelism adopts FlashAttention to improve its processing of long sequences, both RingAttention and BurstAttention have better training time than tensor parallelism when processing long sequences. This is also why existing works using tensor parallelism to train LLMs usually set the training length between 2048 and 4096. Compared with

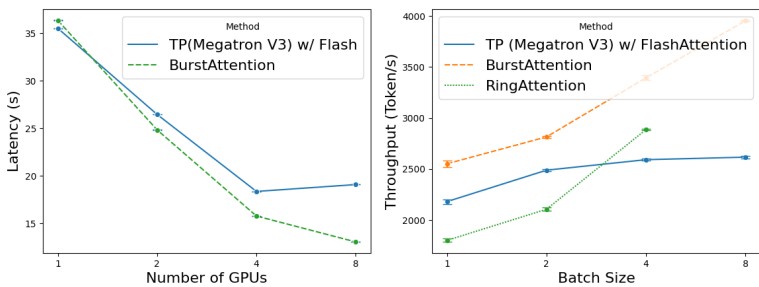

(a) LLaMA-13b latency - GPU number  (b) LLaMA-7b throughput - batch size

Figure 4: Scaling abilities on different GPU numbers and batch sizes.

BurstAttention, RingAttention is limited by the sequence length since it stores too many intermediate states, but BurstAttention can support the longest input length. On the other hand, BurstAttention without LAO has a similar trend of training time as RingAttention and tensor parallelism.

From Figure 3, BurstAttention achieves nearly $2.0\times$ speedup when the sequence is longer than 128K. Also combining BurstAttention with ZeRO optimization brings significant improvements in memory efficiency. Although BurstAttention+ZeRO brings little additional communication overheads, BurstAttention+ZeRO still achieves memory efficiency comparable to Megatron V3 and demonstrates superior speed in both multi-node and single-node setups than Megatron V3. This suggests that BurstAttention, with its current optimizations, offers a more efficient solution in terms of speed, even when faced with a memory-efficient competitor like Megatron V3.

## 5.4 SCALING ABILITY

In this section, we further verify the scaling ability of BurstAttention. In Figure 4a, we set batch size to 1 and sequence length to 65,536, and then evaluate the latency changes with increasing GPU numbers. As shown in the figure, in the single-GPU scenario, BurstAttention with LAO is equivalent to FlashAttention, and its inference latency is on par with the baseline using FlashAttention. Tensor parallelism cannot further decrease the latency when the number of GPUs increases from 4 to 8 due to the communication overhead with increased batch-size, while BurstAttention can achieve better scaling trends. Note that RingAttention requires storing $\Theta(\frac{BZN^2}{G})$ memory for each layer, which is extremely large and cannot fit into GPUs even sharded on 8 GPUs. In Figure 4b, we fix the sequence length to 4096 and the number of GPUs to 8 to evaluate the training throughput changes with increasing batch sizes. The experimental results show that BurstAttention can support a larger batch size, and the throughput grows with the increase of batch sizes in training scenario.

## 5.5 PERPLEXITY

We sample 100 examples from C4 (Raffel et al., 2020) and evaluate the perplexity (PPL) of LLaMA-7b implemented based on different distributed attention solutions. By evaluating PPL scores, we can evaluate the correctness of these implementation. From Table 4, we can find BurstAttention would not bring performance penalty, as compared to other distributed attention solutions.

| Method | PPL |
|---|---|
| TP | 9.901 |
| TP w/ FlashAttention | 9.902 |
| RingAttention | 9.904 |
| BurstAttention w/o LAO | 9.901 |
| BurstAttention | 9.901 |

Table 4: LLaMA-7b PPL on C4.

## 6 CONCLUSION

In this work, we present an efficient distributed attention framework named BurstAttention, which can enhance performance in terms of memory consumption and running speed when processing extremely long sequences. When running on a single device, BurstAttention can achieve comparable efficiency to FlashAttention. When running on a distributed cluster, BurstAttention can outperform existing competitive distributed attention solutions, including RingAttention and tensor parallelism. Moreover, the experimental results show that BurstAttention also has greater scaling abilities than existing solutions as increasing devices and batch sizes.

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
