# OpenReview forum: "BurstAttention: An Efficient Distributed Attention Framework for Extremely Long Sequences"
_ICLR.cc/2024/Conference — Submitted to ICLR 2024_

### Official Review · Reviewer_y1yi · 2023-10-24

**Soundness:** 3 good
**Presentation:** 3 good
**Contribution:** 2 fair
**Rating:** 6
**Confidence:** 3

**Summary:**

> **TL;DR:** The proposed BurstAttention algorithm reduces 40% of communication overheads and thus achieves 2x
 speedup during training 32K sequence length. However, the experiments due not include comparisons to some popular distributed algorithms (Data and Pipeline Parallelism). Addressing my concerns (especially W.1. and W.2.) and questions would improve my score.

The paper introduces BurstAttention, an efficient distributed attention framework designed to address the challenges associated with processing long sequences in Transformer-based large language models (LLMs). While attention modules have been essential for the success of LLMs, their quadratic time and memory complexities pose obstacles for long sequences. BurstAttention divides long sequences into partitions across distributed clusters and employs global and local attention optimization strategies to optimize memory access and communication operations. FlashAttention can be viewed as a specialization of BurstAttention on a single device. Experimental results reveal that BurstAttention reduces communication overheads by 40% and achieves a 2× speedup in training on sequences of 32K length with 8×A100 GPUs, outperforming existing competitive distributed attention solutions. BurstAttention is also shown to adapt well to various optimization strategies and offers greater scalability in comparison to other solutions.

**Strengths:**

* **S.1.** The proposed BurstAttention algorithm tackles an important problem in the computational costs of training and inference of LLMs.
* **S.2.** The paper is well written, the illustrations are informative, and the algorithms are easy to follow.
* **S.3.** The experimental results show that the  BurstAttention algorithm outperforms existing algorithms, especially for long context windows.
* **S.4.** The experiments are conducted on several model sizes and include ablations for the speedup gains.

**Weaknesses:**

* **W.1.** The paper provides comparison to several algorithm, but does not include the common DataParallel and Pipeline Parallelism due to sequence length limitations. As these algorithms are the most popular approaches it would help to show even a single experiments using them as a baseline.
* **W.2.** The experiments are conducted on a single hardware setup, providing experiments on different configurations would help.
* **W.3.** The paper states that BurstAttention can be easily extended to other cross-attention modules, however, it does not provide any details. Providing additional details would help.

**Questions:**

* **Q.1.** Which PCIe is used in the experiments and what is the bandwidth?
* **Q.2.** How would the experiments look if the communication bandwidth would be higher/lower?

---

> ### Author Response · Authors · 2023-11-23
> **Response to your questions**
>
> Thank you for your constructive feedback on our submission. We appreciate your recognition of the strengths of our work and the areas where improvements can be made. We have carefully considered your comments and would like to address your concerns and questions as follows.
>
> ### 1. **For Data/Pipeline Parallelism Baselines**:
>
> We acknowledge the absence of comparisons with common parallelism approaches like data parallelism and pipeline parallelism in our current studies. Our decision to omit these from the initial set of experiments is due to inherent limitations with sequence lengths. From our pilot experiments, directly adopting data parallelism and pipeline parallelism can only handle those sequences shorter than 8192.
>
> We understand the importance of these popular approaches in the field and are planning to conduct further experiments. We aim to explore the applicability of sequence parallelism in various scenarios and compare the differences between different parallelism strategies. This will be a significant aspect of our ongoing research. In our additional experiments for reviewer Gb5f & J3Ya, we have explored combining ZeRO (one optimization for data parallelism) and BurstAttention, and this combination has shown some promising results to trade off runtime and memory efficiency.
>
> ### 2. **Experiments on More Hardware Configurations**:
>
> Frankly, the current experiments are conducted on a specific hardware setup due to the limited resources we can use for this paper. Inspired by your suggestions, we have recognized the value of evaluating our method across different configurations.
>
> However, because of some force majeure, we may need time to prepare other hardware configurations (e.g., 4090 and H100). If these hardware configurations are met, our future research plans include conducting experiments on clusters and devices with more extreme lower/higher communication constraints. We believe this will provide us with a deeper understanding of how BurstAttention performs under varied conditions and will help us in tailoring our attention algorithms and parallelism strategies to more specialized scenarios.
>
> ### 3. **Questions Regarding PCIe and Communication Bandwidth**:
>
> Regarding your first question, our experiments utilize PCIe 4.0, and we do not enable CUDA direct access between GPUs for experiments. The reason for this setting can refer to our additional experiments for reviewer Gb5f & J3Ya.
>
> Concerning your second question, our current experimental results indicate a clear advantage of our approach in scenarios with limited communication capabilities. Since one great advantage of BurstAttention is its low communication overhead and highly overlapping communication and computation operations, the benefits of BurstAttention are less pronounced in environments with high communication bandwidth (e.g., using NVLink and InfiniBand).
>
> Although enabling NVLink and InfiniBand is expensive and such device conditions are uncommon for LLM inference, we have recognized the potential for further optimization when communication bandwidth is higher. We are exploring various communication scenarios to better understand and optimize the performance of BurstAttention under different communication conditions. Thanks again for your insightful suggestion.
>
> We hope these responses address your concerns and questions. We are committed to continuously improving our work and value your input as it guides our future research directions.

---

### Official Review · Reviewer_J3Ya · 2023-10-29

**Soundness:** 3 good
**Presentation:** 3 good
**Contribution:** 2 fair
**Rating:** 5
**Confidence:** 2

**Summary:**

The paper presents BurstAttention, an efficient distributed attention for long context LLMs.

**Strengths:**

1. the method is well motivated, well presented and easy to understand.
2. the comparison is throughout and complete.

**Weaknesses:**

1. The comparison with Megatron is not up-to-date. For instance, Figure 2b does not use the most memory-efficient implementation. The baseline should be set up Megatron-v3 (a combination of tensor model parallelism and sequence parallelism).
2. The latency comparison against Megatron is not well analyzed, see questions below.
3. The argument with sparse attention is vague. There are no experiments.

**Questions:**

Why table 3 shows such a big improvement against Megatron, are they all resulted from the communication volume? In particular, can the author provides the ablation on the communication wall clock time? (1) the communication volume is less indicative, because of different distributed prototype has different throughput themself. For instance, Megatron uses all-gather, which is highly optimized in NCCL. If the system uses P2P (and is the system using P2P? Can the author provide more details?), even the communication volume is less, the wall clock time can be higher. (2) In the experiment, the objective is causal language modeling, where the system seems to have unbalanced workload, e.g. the machine hosting first query will be idle most of the time. However, in Megatron, the workload is balanced because they do not partition sequences. Does the system balance the workload? If not, the system should be 2x slower than Megatron due to the unbalanced workload?

The reviewer is happy to raise score, if these questions are addressed clearly.

---

> ### Author Response · Authors · 2023-11-22
> **Response to your questions(Part 1)**
>
> Thank you for your thoughtful and detailed review of our submission. We appreciate the time you took to analyze our work and the insightful questions you raised. Here are our responses to your questions.
>
> ### 1. **For Comparison with Megatron V3**
>
> Thanks for your insightful suggestion, we have conducted new experiments for comparison with Megatron V3. For the new experiments, we replace the original tensor parallelism with the implementation of Megatron V3, an implementation of tensor parallelism that considers the features of sequence parallelism.
>
> Furthermore, we have expanded our training experiments on 32 A100 GPUs. All these experiments are conducted on a cluster interconnected by a RoCE (RDMA over Converged Ethernet) network with an approximate bandwidth of 200 Gb/s, and each machine of the cluster has eight A100 GPUs connected via PCIe 4.0 with peer-to-peer (P2P) communication disabled. The results of the new experiments are shown in the tables below.
>
> ### 1.1 The Results of Training LLMs
>
> The results below indicate that Megatron V3's tensor parallelism shows better memory efficiency compared to BurstAttention ( since BurstAttention does not shard model parameters ), while BurstAttention has shorter training runtime and lower latency. Moreover, the longer the sequence length is, the more time-efficient BurstAttention is.
>
> Furthermore, combining BurstAttention with ZeRO optimization (ZeRO has very limited benefits over TP since TP itself shards model parameters) brings significant improvements in memory efficiency. Although BurstAttention+ZeRO brings little additional communication overheads, BurstAttention+ZeRO still achieves memory efficiency comparable to Megatron V3 and demonstrates superior speed in both multi-node setups than Megatron V3. This suggests that BurstAttention, with its current optimizations, offers a more efficient solution in terms of speed, even when faced with a memory-efficient competitor like Megatron V3.
>
> #### LLaMA-7b Training Runtime (seconds) on 8xA100 GPUs
>
> | Sequence Length                            | 1024      | 2048      | 4096      | 8192      | 16384     | 32768      |
> |:----------------------------------|:----------|:----------|:----------|:----------|:----------|:-----------|
> | Ring Self-Attention                     | 1.16±0.07 | 0.93±0.03 | 1.49±0.02 | 2.92±0.01 | OOM       | OOM        |
> | TP (Megatron V3) w/ FlashAttention | 1.63±0.11 | 1.37±0.03 | 2.15±0.15 | 3.38±0.08 | 6.93±0.13 | 14.76±0.06 |
> | BurstAttention                    | 1.05±0.09 | 0.77±0.03 | 1.62±0.05 | 3.08±0.06 | 5.18±0.15 | 9.93±0.02  |
>
> #### LLaMA-7b Training Memory Usage (GB) on 8xA100 GPUs
>
> | Sequence Length                            |   1024 |   2048 |   4096 |   8192 | 16384   | 32768   |
> |:----------------------------------|-------:|-------:|-------:|-------:|:--------|:--------|
> | Ring Self-Attention                     |  26.94 |  28.54 |  33.2  |  48.97 | OOM     | OOM     |
> | TP (Megatron V3) w/ FlashAttention |   5.81 |   6.66 |   8.35 |  11.75 | 18.51   | 32.11   |
> | BurstAttention                    |  26.69 |  27.55 |  29.23 |  32.93 | 40.26   | 54.69   |
>
> #### LLaMA-7b Training Runtime (seconds) on 32xA100 GPUs
>
> | Sequence Length                            | 32768      | 65536      | 98304      | 131072     |
> |:----------------------------------|:-----------|:-----------|:-----------|:-----------|
> | TP (Megatron V3) w/ FlashAttention | 17.14±0.10 | 34.95±0.18 | 55.35±0.46 | 73.99±0.23 |
> | BurstAttention                    | 8.93±0.03  | 18.12±0.13 | 28.91±0.05 | 40.61±0.13 |
> | BurstAttention+ZeRO               | 13.81±0.07 | 23.12±0.09 | 33.56±0.07 | 44.22±0.06 |
>
>
> #### LLaMA-7b Training Memory Usage (GB) on 32xA100 GPUs
>
> | Sequence Length                            |   32768 |   65536 |   98304 |   131072 |
> |:----------------------------------|--------:|--------:|--------:|---------:|
> | TP (Megatron V3) w/ FlashAttention |    9.89±0.01 |   18.17±0.01 |   26.43±0.01 |    34.70±0.01  |
> | BurstAttention                    |   35.87±0.01 |   46.15±0.01 |   56.24±0.01 |    66.45±0.01 |
> | BurstAttention+ZeRO               |   11.55±0.01 |   21.83±0.01 |   31.92±0.01 |    42.14±0.01 |
>
> ### 1.2 The Results of Inferring LLMs
>
> Besides the results of training LLMs, we also show the results of inferring LLMs. In terms of inferring LLMs, BurstAttention still achieves a speed improvement than Megatron V3. Note that although TP (Megatron V3) is more memory efficient than TP (Megatron V1), the all-reduce operation used by TP (Megatron V1) is better optimized than the reduce-scatter and all-gather operations used by TP (Megatron V3). In the actual inference process, TP (Megatron V1) is even slightly faster than TP (Megatron V3).
>
>
> [To be continued]

---

> ### Author Response · Authors · 2023-11-22
> **Response to your questions(Part 2)**
>
> #### LLaMA-7b Inference First Token Latency (seconds) on 8xA100 GPUs
> | Sequence Length                 | 4096      | 8192      | 16384     | 32768     | 65536      | 131072     | 262144     |
> |:-----------------------|:----------|:----------|:----------|:----------|:-----------|:-----------|:-----------|
> | Ring Self-Attention    | 0.42±0.01 | 0.87±0.01 | 2.00±0.01 | 5.13±0.05 | OOM        | OOM        | OOM        |
> | TP (Megatron V1) w/ Flash   | 0.67±0.01 | 1.29±0.01 | 2.58±0.01 | 5.27±0.01 | 11.63±0.02 | 27.54±0.01 | 71.52±0.06 |
> | TP (Megatron V3) w/ Flash   | 0.73±0.02 | 1.36±0.01 | 2.68±0.01 | 5.67±0.01 | 12.25±0.01 | 28.73±0.03 | 75.52±0.05 |
> | BurstAttention w/o LAO | 0.46±0.01 | 0.88±0.01 | 1.79±0.01 | 3.88±0.01 | 10.78±0.01 | OOM        | OOM        |
> | BurstAttention         | 0.44±0.01 | 0.84±0.01 | 1.68±0.01 | 3.27±0.01 | 6.49±0.01  | 16.01±0.01 | 49.32±0.11 |
>
> #### LLama-13b Inference First Token Latency (seconds) on 8xA100 GPUs
> | Sequence Length                 | 4096      | 8192      | 16384     | 32768     | 65536      | 131072     | 262144      |
> |:-----------------------|:----------|:----------|:----------|:----------|:-----------|:-----------|:------------|
> | Ring Self-Attention    | 0.66±0.01 | 1.36±0.01 | 3.08±0.01 | 7.98±0.02 | OOM        | OOM        | OOM         |
> | TP (Megatron V1) w/ Flash   | 1.05±0.01 | 2.01±0.01 | 4.03±0.01 | 8.41±0.01 | 18.56±0.02 | 44.39±0.04 | OOM         |
> | TP (Megatron V3) w/ Flash   | 1.07±0.01 | 2.09±0.01 | 4.20±0.01 | 8.76±0.01 | 19.06±0.06 | 45.46±0.03 | 119.03±0.04 |
> | BurstAttention w/o LAO | 0.72±0.01 | 1.39±0.01 | 2.77±0.05 | 5.99±0.01 | 16.95±0.01 | OOM        | OOM         |
> | BurstAttention         | 0.69±0.01 | 1.40±0.05 | 2.57±0.03 | 5.08±0.02 | 9.92±0.01  | 25.91±0.01 | 78.80±0.07  |
>
>
> ### 1.3 The Results of Scaling Experiments
>
> Some reviewers suggest that we should provide some experimental results of scaling GPU number, sequence length, and batch size. We list the results of scaling experiments in the tables below.
>
> #### LLama-13b Inference First Token Latency (seconds) with Scaling the GPU Number and Sequence Length.
>
>
> **Part 1: Performance on 1x and 2x GPU Configurations**
>
> | Setting                   | 1xGPUs SeqLen 16384   | 1xGPUs SeqLen 32768   | 1xGPUs SeqLen 65536   | 2xGPUs SeqLen 16384   | 2xGPUs SeqLen 32768   | 2xGPUs SeqLen 65536   |
> |:-------------------------|:----------------------|:----------------------|:----------------------|:----------------------|:----------------------|:----------------------|
> | TP (Megatron V3) w/ Flash | 3.70±0.01             | 10.90±0.01            | 35.52±0.01            | 4.01±0.01             | 9.85±0.01             | 26.45±0.01            |
> | BurstAttention           | 3.86±0.01             | 11.23±0.01            | 36.35±0.02            | 3.52±0.01             | 8.93±0.01             | 24.83±0.01            |
>
> **Part 2: Performance on 4x and 8x GPU Configurations**
>
> | Setting                   | 4xGPUs SeqLen 16384   | 4xGPUs SeqLen 32768   | 4xGPUs SeqLen 65536   | 8xGPUs SeqLen 16384   | 8xGPUs SeqLen 32768   | 8xGPUs SeqLen 65536   |
> |:-------------------------|:----------------------|:----------------------|:----------------------|:----------------------|:----------------------|:----------------------|
> | TP (Megatron V3) w/ Flash | 3.37±0.01             | 7.52±0.01             | 18.36±0.01            | 4.13±0.01             | 8.88±0.01             | 19.08±0.01            |
> | BurstAttention           | 2.66±0.01             | 6.06±0.01             | 15.77±0.01            | 2.73±0.01             | 5.77±0.01             | 13.05±0.01            |
>
>
> #### LLaMA-7b Training Throughput (Token/s) with Scaling the Batch Size
>
> | Setting                 | Batch 1 Seqlen 4096   | Batch 2 Seqlen 4096   | Batch 4 Seqlen 4096   | Batch 8 Seqlen 4096   |
> |:-----------------------|:----------------------|:----------------------|:----------------------|:----------------------|
> | TP (Megatron V3) w/ FlashAttention | 2180.98±23.42 | 2489.12±7.60  | 2592.22±8.21  | 2616.87±12.61 |
> | BurstAttention         | 2551.82±32.24 | 2815.29±9.73  | 3395.43±21.25 | 3955.27±4.80  |
> | RingAttention          | 1800.74±15.99 | 2106.35±16.37 | 2887.44±5.11  | OOM       |
>
> The scaling experiment results proved that BurstAttention can be more efficient in both training and inferencing.
>
> For inferencing, BurstAttention improves its performance as the number of GPUs and sequence length increase. This suggests that BurstAttention scales effectively across different hardware configurations and can handle larger sequences efficiently. Also, it demonstrates lower first token latency across various sequence lengths compared to TP (Megatron V1 and V3) with FlashAttention and Ring Self-Attention.
>
> [To be continued]

---

> ### Author Response · Authors · 2023-11-22
> **Response to your questions(Part 3)**
>
> For training, the results of the LLaMA-7b model training experiments indicate that BurstAttention has higher training throughput compared to the tensor parallelism of Megatron V3 with FlashAttention, especially as the batch size increases. This points to BurstAttention's capability to process larger batches more efficiently, which is crucial for training LLMs effectively.
>
> While BurstAttention exhibits impressive performance in various aspects, it's important to acknowledge certain limitations, as highlighted by experimental results. Specifically, it has been observed that in certain scenarios, Megatron V3's tensor parallelism demonstrates a more efficient memory footprint compared to BurstAttention. It's noteworthy that when integrated with the ZeRO (Zero Redundancy Optimizer) technique, BurstAttention not only achieves comparable efficiency but can also surpass Megatron in terms of throughput. This integration effectively leverages the strengths of both technologies, optimizing memory usage while enhancing overall processing speed.
>
>
> ### 2. **For Peer-to-Peer (P2P) Communication settings**
>
> Firstly, we don't know that the reviewer is referring to the P2P communication operation such as send-recv or the P2P transport in the PCIe network, which means CUDA direct access between GPUs. We will presume that the reviewer is referring to CUDA direct access between GPUs before we continue the following discussion. However, if the reviewer's focus is indeed on the communication operations such as send-recv, then it should be noted the ring-collective operations are implemented using send-recv operation.
>
> We acknowledge your observation regarding our experiments' absence of peer-to-peer (P2P) communication. Since we focus on proposing a distributed attention framework that solves extremely long sequences, it is hard for communication across machines to utilize P2P features. Moreover, it is also difficult for the environments of inferring LLMs (often using NVIDIA 4090 or L40 for inference) to meet P2P conditions. Hence, in our experiments, we turn off P2P communication as it is not a common setting in many communication-restricted scenarios (e.g., NVIDIA 3090/4090, NVIDIA A100/V100 PCIe, other non-NVIDIA devices, and the scenarios mentioned above).
>
> In our latest experiments, we have optimized the original BurstAttention framework by overlapping the computation and communication phases. As a result, we can observe an acceleration in performance, evident in both multi-node and single-node experiments. Intriguingly, this optimization appears to be particularly efficacious in the variant of BurstAttention without FlashAttention, which is inherently more computation-intensive. This observation suggests a significant interplay between computational density and the effectiveness of our communication-computation overlapping strategy, opening avenues for further exploration in the optimization of distributed attention mechanisms.
>
>
> ### 3. **For the Workload Balancing of Causal Language Modeling**:
>
> Regarding your query about workload balancing, we have not optimized the causal language modeling process in BurstAttention or other experiments. In our current implementation, we adhere to the conventional attention-masking approach without skipping computations. Hence, for both BurstAttention and Megatron V1/V3, the workload is balanced, and the experimental comparison is fair.
>
> In fact, adopting the conventional attention-masking approach for parallel computing brings unnecessary computational overheads. For future work, we aim to address the workload balancing issue of causal language modeling by exploring more complex communication topologies beyond simple ring communication. A potential strategy we are considering involves repartitioning the sequence dimension across those dimensions unrelated to computation to ensure an even distribution of the attention mask across devices.
>
> ### 4. **For the Argument with Sparse Attention**:
>
>
> We acknowledge your observation that the sparse attention aspect of our work may appear vague. Since implementing sparse attention based on BurstAttention just requires changing the attention masking matrix, we thus only briefly introduce the compatibility of BurstAttention with sparse attention.
>
> This question is very insightful, especially as it highly relates to the workload imbalance issue. We are still working on addressing the workload imbalance issue associated with causal language modeling, and the imbalance issue is more serious for sparse attention. We are developing more refined strategies to effectively manage this challenge, providing greater clarity and depth in our future iterations.
>
> Thanks again for your suggestion, and we will add more discussions about the workload imbalance issue and sparse attention in our revision, which will make our work more solid.

---

> > ### Comment · Reviewer_J3Ya · 2023-11-22
> >
> > Thank you for getting back to the important questions, with such a detailed analysis.
> >
> > I should be more clear on P2P, I actually meant the send/recv in the communication operation. I am bringing up this because these operations are not optimized by themselves. Concretely, if you are using implementing BurstAttention by send/recv, you will likely miss a lot of optimizations in all-gather-like kernels (e.g. topology-aware). This is also pointed out in the RSA paper. This may be less of an issue if you implement overlapping in a good way.
> >
> > For the second problem, I don't think my concerns are addressed. Megatron-LM can use flash-attention with causal implementation in the kernel, and it is not fair for them to use an "attention-masking" approach. In BurstAttention, you are manipulating in the sequence dimension, and are responsible to recover the causal implementation yourself. This is a key design, vs tensor-parallelism and sequence parallelism over attention.
> >
> > The review can not raise score mainly due to the second reason. However, I will reduce my confidence score, and let ACs to judge the significance of the issue.

---

> > > ### Author Response · Authors · 2023-11-23
> > > **Futher explanation for the second problem**
> > >
> > > In the context of multi-device distributed training, we have overlapped communication and computation. This overlapping is especially pertinent in communication-constrained scenarios, which are our primary focus. In such scenarios, even though BurstAttention has an unbalanced computation workload, all the computation is overlapped in communication, making communication the bottleneck. In contrast, although Megatron has a balanced computation workload, their computation needs to be conducted after communication. Given that BurstAttention has a much lower communication volume compared to Megatron TP, it is more practical in scenarios with extremely long sequence lengths or in communication-constrained environments.
> > >
> > > On the other hand, to achieve load balancing, we can adjust the sequence sharding strategy using the attention head dimension. For example, in a 2-GPU scenario, instead of letting the first GPU maintain vectors of [0..B, 0..H, 0..(L/2), 0..D] and letting the second GPU maintain vectors of [0..B, 0..H, (L/2)..L, 0..D], we can let the first GPU maintain [0..B, 0..(H/2), 0..(L/2), 0..D] and [0..B, (H/2)..H, (L/2)..L, 0..D], and let the second GPU maintain [0..B, 0..(H/2), (L/2)..L, 0..D] and [0..B, (H/2)..H, 0..(L/2), 0..D]. Through this cyclic shifting, load balancing across multiple devices in distributed training can be achieved. This will be left for our future work.

---

> > > > ### Comment · Area_Chair_fwfb · 2023-12-05
> > > >
> > > > Dear Reviewer J3Ya,
> > > >
> > > > Please help to read the author's response and make decision. Thanks.
> > > >
> > > > Bests, AC

---

### Official Review · Reviewer_Jngs · 2023-11-01

**Soundness:** 3 good
**Presentation:** 3 good
**Contribution:** 2 fair
**Rating:** 6
**Confidence:** 4

**Summary:**

The paper aims at processing extremely long sequences in attention under distributed settings. The proposed solution is to partition the computation of attention modules across multiple devices at the sequence dimension, which seems to be an improved version of the prior RingAttention method. Through the proposed global attention optimization using online softmax, the BurstAttention method can reduce the memory usage of intermediate results. In addition, BurstAttention can be integrated with FlashAttention, as the local attention optimization, to further reduce latency and support longer sequences.

**Strengths:**

- The paper is well-motivated at addressing the memory and communication challenges of processing extremely long sequences in multi-device settings.
- The proposed global attention optimization seems to be an effective extension to RingAttention in reducing memory usage.

**Weaknesses:**

While integrating two prior methods (RingAttention and FlashAttention) to be more efficient by itself could be a contribution, the paper could elaborate more on the new challenges, if any, and different method designs to establish its new contributions. In particular, the local attention optimization does not seem to bring any new insights than using FlashAttention in a distributed system.

**Questions:**

- Can you comment on the significance of the reduced latency of the first token? How is the implication on overall latency?
- What is the new contribution over FlashAttention? Or simply integrating it into the BurstAttention framework?

---

> ### Author Response · Authors · 2023-11-22
> **Response to regarding the two questions**
>
> Thank you for your insightful review and valuable questions regarding our paper. We appreciate your recognition of our work's motivation and effectiveness in addressing the challenges of processing extremely long sequences in distributed settings. We would like to address your questions and concerns as follows.
>
> ### 1. **The Significance of Reducing the First Token Latency for Inference**:
>
> Although the first token latency might represent a small fraction of the overall inference process, it is critically important for real-time AI services such as ChatGPT. In such applications, the system's responsiveness significantly impacts the user experience, and these applications usually output results in a streaming manner to improve responsiveness. Since the first token's latency is the longest, reducing the first token's latency is thus a crucial performance metric, as it directly influences the perceived responsiveness and efficiency of the model in these user-facing streaming scenarios. [1] and [2] also explain why the first token latency is an important metric for LLM serving, which can provide more details.
>
> [1] https://www.databricks.com/blog/llm-inference-performance-engineering-best-practices
> [2] https://awsdocs-neuron.readthedocs-hosted.com/en/latest/general/appnotes/transformers-neuronx/generative-llm-inference-with-neuron.html
>
> ### 2. **The Contributions of BurstAttention over RingAttention and FlashAttention**:
>
> Our main contribution is to build a unified distributed framework for extremely-long sequence parallelism. Under the unified framework, by comprehensively considering the characteristics of clusters and devices, we propose local attention optimization to reduce memory overhead, global attention optimization to reduce communication overhead, and further overlap of global operations and local operations to minimize latency. Rather than an integration of global attention operations and local attention operations, BurstAttention is a significant reimagining and enhancement of how devices interact and function together in a distributed system to perform attention for long sequences.
>
> Although we mentioned in the paper that RingAttention and FlashAttention can be seen as the degradation of BurstAttention in specific scenarios, this cannot simply be seen as BurstAttention being a combination of global RingAttention and local FlashAttention. In fact, since RingAttention will first compute all local attention scores and then conduct softmax operations across machines to obtain global results, this attention mode makes it difficult to cooperate with the online softmax operations of FlashAttention. This means that combining RingAttention and FlashAttention is non-trivial. Even if RingAttention and FlashAttention are combined, this simple combination cannot fully utilize the cluster and device characteristics for performance optimization.
>
> In summary, the unified framework and the derived optimization strategies are central to our contribution, enabling us to specifically optimize the ring-collective operations, fully overlap computation and communication, and process much longer sequences more efficiently than existing baselines. We will add more discussion of this in our revision.
>
>
> Thanks again for your suggestion. We hope these responses will clarify the aspects highlighted in your review. We value your feedback and think these suggestions can significantly help us improve our work and its presentation.

---

### Official Review · Reviewer_s62K · 2023-11-01

**Soundness:** 3 good
**Presentation:** 3 good
**Contribution:** 2 fair
**Rating:** 6
**Confidence:** 3

**Summary:**

The paper contributes a new sequence parallelism scheme for attending over long sequences that uses GPU memory hierarchy and distributed ring communication to improve inference and training throughput. The authors show the effectiveness of this method on Llama2-7B for training and inference and Llama2-13B for inference, and show improvements over both TP+FlashAttention, which has lower memory footprint but lower throughput, and RingAttention, which has higher memory footprint and higher throughput. The authors also show that their formulation, which uses local and global attention optimization to improve memory footprint and global communication, maintains exact attention correctness while still bringing these performance improvements.

**Strengths:**

The paper analysis for training and inference time tradeoff with respect to memory and throughput is solid for causal decoder-only transformer models. The algorithm and communication and memory overhead analysis is also very detailed and thorough for both GAO and LAO.

**Weaknesses:**

The paper only evaluates two decoder-only models at inference time, and a single small model at training time. The scaling law along the sequence dimension is present but a scaling law along the model capacity dimension is also important given that the dimensionality of the Q/K/V will also increase with model size and therefore the communication and memory overhead as well.

**Questions:**

For the tensor parallelism/TP+Flash Attention baselines, is the implementation being used the Megatron-LM implementation which parallelizes each head across devices (and therefore reduces the memory footprint/communication)?

For the time-to-first-token experiment is the sequence length equivalent to the prompt input length e.g. the model generates the first token conditioned on 4096/8192/.../262144 tokens?

---

> ### Author Response · Authors · 2023-11-22
> **Response to your question**
>
> Thank you for your insightful feedback on our submission. We greatly appreciate your recognition of the thorough analysis and detailed algorithmic contributions in our paper. Below, we address each of your concerns and questions to clarify our approach and its implications.
>
> ### 1. **For the Details of Tensor Parallelism Implementation**:
>
> Yes, we use the tensor parallelism implementation of Megatron for our experiments, which parallelizes attention heads across devices to reduce the memory footprint. However, even with head-wise parallelization, Megatron V1's tensor parallelism still exhibits substantial memory usage. Notably, the communication volume in Megatron V1's attention mechanism remains significant, particularly the all-reduce operation to obtain attention outputs. Hence, the communication and memory overheads are not notably reduced by parallelizing along the attention head dimension. More importantly, the increase of memory overhead brought by parameter growth is linear, while the rise brought by sequence length growth is quadratic, i.e., the core memory overhead is to store attention intermediate results rather than storing parameters in processing extremely long sequences, which cannot be solved by only parallelizing attention heads.
>
> In supplementary experiments, we compare our method with the tensor parallelism approach in Megatron V3, which employs all-gather and reduce-scatter operations instead of all-reduce operations in Megatron V1. For Megatron V3's approach, tensor parallelism's communication volume remains akin to that of Megatron V1, but the memory footprint is significantly reduced. These supplementary results can refer to the responses to Reviewer Gb5f and Reviewer J3Ya.
>
> ### 2. **For the First Token Latency**:
>
> Regarding your question on the first token latency, you are correct. This metric indeed refers to the time taken for the first inference step, specifically the encoding time. It is an essential measure since the computation for long sequence attention primarily occurs during the encoding phase in inference. In real-time AI services such as ChatGPT, the system's responsiveness significantly impacts the user experience, and these applications usually output results in a streaming manner to improve responsiveness. Since the first token's latency is the longest, reducing the first token's latency is thus a crucial performance metric, as it directly influences the perceived responsiveness and efficiency of the model in these streaming scenarios. [1] and [2] also explain why the first token latency is an important metric for LLM serving, which can provide more details.
>
> [1] https://www.databricks.com/blog/llm-inference-performance-engineering-best-practices
> [2] https://awsdocs-neuron.readthedocs-hosted.com/en/latest/general/appnotes/transformers-neuronx/generative-llm-inference-with-neuron.html
>
> ### 3. **For the Scaling Law Along the Model Capacity Dimension**:
>
> We acknowledge the importance of evaluating the scaling law along the model capacity dimension. In this paper, our analysis primarily focuses on the O(N^2) attention complexity and its optimization through BurstAttention. To clearly evaluate the effectiveness and efficiency of different methods to handle extremely long attention, we choose a fixed model size for our experiments, which can bring an optimization perspective that is independent of the trade-offs introduced by parallel methods and optimizations.
>
> We understand that experiments on larger models would require additional optimization strategies, and the performance bottlenecks may vary based on different hardware configurations. Therefore, simply comparing the performance on larger models and clusters without these considerations may not yield clear insights. We agree this suggestion is meaningful and plan to extend our analysis to larger models in future work, taking into account these complex optimization and device factors.

---

> > ### Comment · Reviewer_s62K · 2023-11-22
> >
> > Thank you for the detailed response. The clarification on these points is helpful. Without analysis on larger models, I still have concerns that this method may not scale as suggested in the paper. I will maintain my score.

---

> > > ### Author Response · Authors · 2023-11-23
> > >
> > > Thank you for your insights. Indeed, our current hardware resources are limited, which constrains our ability to conduct experiments on larger models. So far, we have managed to train the LLaMA-70b model using a 32xA100 GPU cluster, with enable checkpointing activation. We plan to provide detailed results of the LLaMA-70b experiments in the future as our resources allow.

---

> > > > ### Author Response · Authors · 2023-11-23
> > > > **Additional Experiments results**
> > > >
> > > > We have already finished BurstAttention scaling experiments on LLaMa-70b  training. This experiments conducted on a cluster connected with 600Gb/s RoCE network and each machine have 8xA100 connected with nvlink.
> > > > # LLaMa-70b Training Experiments on 32xA100 GPUs
> > > > | Method                            | 32768      | 65536      | 131072      |
> > > > |:----------------------------------|:-----------|:-----------|:------------|
> > > > | BurstAttention+ZeRO               | 36.05±0.59 | 69.84±0.41 | 154.31±0.36 |
> > > >
> > > >
> > > > Based on the scaling results with sequece length, we can see the training time actually increases linearly.
> > > > We haven't finished the TP(Megatron V3) experiments  so far and we would provide experiments results in the future.

---

### Official Review · Reviewer_Gb5f · 2023-11-02

**Soundness:** 2 fair
**Presentation:** 3 good
**Contribution:** 2 fair
**Rating:** 5
**Confidence:** 4

**Summary:**

The paper presents BurstAttention, a method for partitioning the attention operation across multiple GPUs in order to process very long sequences. The method combines a global step, which computes attention using a ring-based method and online softmax, with a blocked local attention implementation to take advantage of on-chip cache. The memory, local I/O, and communication costs of BurstAttention are analyzed. Finally, BurstAttention is evaluated experimentally on LLaMA-2 7- and 13-billion parameter models for inference and training performance, where BurstAttention outperforms other methods such as RingAttention.

**Strengths:**

1. Long sequence lengths are a critical problem for LLMs, and sparse attention mechanisms may not always be sufficient to tackle them. This paper is helping to address this, both in inference and training, by providing a method to decompose the attention computation among multiple GPUs.
2. The combination of the global and local attention optimizations, plus online softmax, enable improved performance in the paper's benchmark results.
3. The benchmark results compare with several existing methods for high-performance attention computation, including FlashAttention and RingAttention.

**Weaknesses:**

1. The novelty of the paper is unclear. It seems to me that it is essentially a straightforward combination of three existing ideas: RingAttention (Li et al., 2022); online softmax (Milakov & Gimelshein, 2018); and FlashAttention (Dao et al., 2022). Further, the ring-based approach to attention seems essentially to be a variant on classic 1D distributed matrix-matrix multiplication algorithms or Cannon's algorithm with one dimension of the mesh trivial (see, e.g., Golub's "Matrix Computations" book, or the Cannon or Fox algorithm). One could imagine pipelining the distributed attention computation (it is unclear to me whether this paper does this), but then it still seems very similar to SUMMA (Van De Geijn & Watts, 1997).
2. The analysis section seems quite perfunctory. It adequately describes the characteristics of the particular implementation, but does not address optimality. E.g., there are lower bounds for distributed matrix-matrix multiplication (via pebbling and in communication-avoiding algorithms) that could be used. Similarly, FlashAttention includes a lower-bound analysis that might be leveraged here.
3. The experimental evaluation does not include any error bars or measures of variance. Especially since this includes cross-device communication, and BurstAttention introduces additional communication on top of existing methods (e.g., tensor or data parallelism) which may interfere, these are important to include.
4. The experimental section lacks any detailed analysis of the algorithm and includes only end-to-end results. For example, what percent of peak device flop was sustained? How much communication is performed? What is the breakdown in runtime of communication and computation? How well does the local attention optimization perform (e.g., reduction in cache misses)? These sorts of analyses would help inform how BatchAttention performs in different scenarios, what its bottlenecks are, and why, precisely, it is faster than existing methods. (Further, the paper states in Section 1 that BurstAttention "can fully optimize the input-output (I/O) and communication procedures in distributed clusters", yet does not back this claim up.)
5. The evaluation is conducted only up to 8 GPUs which are interconnected via PCIe. This is not especially large scale for training LLMs (although could be reasonable for fine-tuning), and it is not clear how the method will scale.

**Questions:**

1. Please clarify the novelty of the paper (see above).
2. Can you contextualize the analysis in Section 4 with lower bounds or otherwise provide an indication of how far BurstAttention is from optimal?
3. Please add error bars (or similar) to all experimental results.
4. Please expand the experimental analysis; see above for suggestions.
5. How does BurstAttention scale to larger numbers of GPUs or on different networks?
6. A related point, how does BurstAttention perform when combined with other standard parallelism techniques for training, e.g., data-parallelism/FSDP, pipelining, etc.? Does its additional communication cause interference?
7. In Section 5.3, why fix the batch size to 1 for training? This seems quite small.
8. In Section 5.4, why is a different sequence length used for scaling than is presented in the earlier results? This makes it hard to contextualize the scaling.

-----

In light of the authors' response addressing some of my concerns, I have raised my score. However, I still remain concerned about the novelty and depth of the experiments.

---

> ### Author Response · Authors · 2023-11-23
> **Response to your questions(Part 1)**
>
> Thank you for your insightful comments and questions regarding our manuscript. Your feedback is invaluable in enhancing the quality and comprehensiveness of our research. We have carefully considered each point you raised and provide the following responses.
>
> ### 1. **For the Novelty**
>
> Our main contribution is to build a unified distributed framework for extremely-long sequence. Under the unified framework, by comprehensively considering the characteristics of clusters and devices, we propose local attention optimization to reduce memory overhead, global attention optimization to reduce communication overhead, and further overlap of global operations and local operations to minimize latency. Rather than an integration of global attention operations and local attention operations, BurstAttention is a significant reimagining and enhancement of how devices interact and function together in a distributed system to perform attention for long sequences.
>
> Although we mentioned in the paper that RingAttention and FlashAttention can be seen as the degradation of BurstAttention in specific scenarios, this cannot simply be seen as BurstAttention being a combination of global RingAttention and local FlashAttention. In fact, since RingAttention will first compute all local attention scores and then conduct softmax operations across machines to obtain global results, this attention mode makes it difficult to cooperate with the online softmax operations of FlashAttention. This means that combining RingAttention and FlashAttention is non-trivial. Even if RingAttention and FlashAttention are combined, this simple combination cannot fully utilize the cluster and device characteristics for performance optimization.
>
> In summary, the unified framework and the derived optimization strategies are central to our contribution, enabling us to specifically optimize the ring-collective operations, fully overlap computation and communication, and process much longer sequences more efficiently than existing baselines. We will add more discussion of this in our revision.
>
>
> ### 2. **For More Experiments and Analysis**
>
> Thanks for your insightful suggestion, we have added some new experiments to make our claim more clear. And we also take the suggestion of Reviewer J3Ya and conduct new experiments for comparison with Megatron V3. For the new experiments, we replace the original tensor parallelism with the implementation of Megatron V3, an implementation of tensor parallelism that considers the features of sequence parallelism. These results can refer to 2.1.
>
> Furthermore, we have expand our training experiments on 32 A100 GPUs. All these experiments are conducted on a cluster interconnected by a RoCE (RDMA over Converged Ethernet) network with an approximate bandwidth of 200 Gb/s, and each machine of the cluster has eight A100 GPUs connected via PCIe 4.0 with peer-to-peer (P2P) communication disabled. The results of the new experiments are shown in the tables below.
>
> #### For Error Bars
>
> Thank you for your insightful suggestion. Our experiments in the paper are the average of multiple experiments, yet we neglect to give error bars as the results of our multiple experiments are very close. We have followed your suggestion to add error bars for both existing experiments and new experiments. Thanks again for your suggestion.
>
> #### For Cluster Scale
>
> We acknowledge your observation regarding the limited scale of the cluster used in our experiments. We agree that evaluating our BurstAttention framework on a larger-scale cluster would be beneficial for further validating its scalability and efficiency with longer sequences. The initial experiments were conducted on a modestly sized cluster to demonstrate the feasibility and efficacy of our approach.
>
> Our expanded experiments on this 32-GPU setup, with the training length extended to 128k, have demonstrated notable improvements in speed over the tensor parallel approach, especially under these specific network and hardware configurations. Even models with hundreds of billions of parameters like GPT-3 rarely use more than 8 GPUs for inference, so this 32-GPU experiments are only conducted on training. These results underscore the robustness and scalability of our method, affirming its effectiveness in a real-world, high-performance computing environment.
>
> [To be continued]

---

> ### Author Response · Authors · 2023-11-23
> **Response to your questions(Part 2)**
>
> #### The Detailed Analysis of Experiments Results
> We sincerely appreciate your valuable suggestion regarding the need for more detailed experimental analysis. We understand that fine-grained analysis can offer deeper insights into the practical bottlenecks of BurstAttention.
>
> However, in the context of multi-device distributed training, we have overlapped communication and computation. This overlapping is especially pertinent in communication-constrained scenarios, which are our primary focus. In such scenarios, computation are often subsumed within communication times, rendering traditional metrics like sustained flop percentage less applicable. Furthermore, our algorithm includes non-matmul FLOPs, which further complicates the use of flop-based metrics for comprehensive performance assessment.
>
> Given these considerations, we opted for end-to-end results as our primary performance indicator. This metric directly reflects the extent of optimization achieved by BurstAttention.
>
> Nevertheless, we understand that fine-grained would help to explain where the bottleneck is and how BurstAttention sloved it. Moving forward, we plan to conduct additional experiments on a variety of cluster configurations and attempt to pinpoint the optimal scenarios for BurstAttention's deployment. This will involve investigating how different input shapes, GPU performances, and communication bandwidths impact the effectiveness of BurstAttention compared to other algorithms. In the course of these experiments, we will endeavor to include more detailed profiling of BurstAttention to provide a clearer understanding of its performance characteristics.
>
> #### For the Combination of BurstAttention and Data Parallelism/FSDP
>
> Since our framework is for both training and inference, and data/pipeline parallelism methods are seldomly adopted for inference, we mainly focus on comparing our framework with existing sequence and tensor parallelism methods. In our new experiments, we combine BurstAttention and ZeRO together to show the effect of combing BurstAttention and Data Parallelism/FSDP, and these results can refer to 2.1.
>
> Note that, combing data/pipeline methods and tensor/sequence parallelism methods only introduces additional communication costs, and has no side effects on our analysis on attention with extremely long sequences.
>
>
>
> #### For the Choice of Batch Size and Sequence Length
>
> For training LLMs, the total tokens of a batch are required to be between 2M and 4M, otherwise, the model performance may be degraded. This means that the longer the training sequence length is, the smaller the batch size is. Due to this, several GPUs may need to compute one example. For example, using 2048 GPUs to train 128-layer GPT-3, the sequence length is 4096, the batch size is 1024, data parallelism is 16, pipeline parallelism is 32, and tensor parallelism is 4. In this scenario, the optimal setup is to divide a batch into 64 micro-batches with a micro-batch size of 1. In this case, four GPUs under the tensor parallelism group are inevitably required to calculate one piece of data together. In view of this, in our paper, we use batch size = 1 for our experiments.
>
> To make our experimental results more sufficient, we follow the reviewers' suggestions and add more experiments with scaling batch and sequence sizes. These results are shown in 2.3.
>
> ### 2.1 The Results of Training LLMs
>
> The results below indicate that while Megatron V3's tensor parallelism shows better memory efficiency compared to BurstAttention, since BurstAttention does not do model parameters' sharding, BurstAttention has shorter training runtime and better time efficiency. Moreover, the longer the sequence length, the more time-efficient BurstAttention is.
>
> Furthermore, combining BurstAttention with ZeRO optimization (ZeRO has very limited benefits over TP since TP itself shards model parameters) brings significant improvements in memory efficiency. Although BurstAttention+ZeRO brings little additional communication overheads, BurstAttention+ZeRO still achieves memory efficiency comparable to Megatron V3 and demonstrates superior speed in both multi-node and single-node setups than Megatron V3. This suggests that BurstAttention, with its current optimizations, offers a more efficient solution in terms of speed, even when faced with a memory-efficient competitor like Megatron V3.
>
>
> [To be continued]

---

> ### Author Response · Authors · 2023-11-23
> **Response to your questions(Part 3)**
>
> #### LLama-7b Training Runtime (seconds) on 8xA100 GPUs
>
> | Sequence Length                            | 1024      | 2048      | 4096      | 8192      | 16384     | 32768      |
> |:----------------------------------|:----------|:----------|:----------|:----------|:----------|:-----------|
> | Ring Self-Attention                     | 1.16±0.07 | 0.93±0.03 | 1.49±0.02 | 2.92±0.01 | OOM       | OOM        |
> | TP(Megatron V3) w/ FlashAttention | 1.63±0.11 | 1.37±0.03 | 2.15±0.15 | 3.38±0.08 | 6.93±0.13 | 14.76±0.06 |
> | BurstAttention                    | 1.05±0.09 | 0.77±0.03 | 1.62±0.05 | 3.08±0.06 | 5.18±0.15 | 9.93±0.02  |
>
> #### LLama-7b Training Memory Usage (GB) on 8xA100 GPUs
>
> | Sequence Length                            |   1024 |   2048 |   4096 |   8192 | 16384   | 32768   |
> |:----------------------------------|-------:|-------:|-------:|-------:|:--------|:--------|
> | Ring Self-Attention                     |  26.94 |  28.54 |  33.2  |  48.97 | OOM     | OOM     |
> | TP(Megatron V3) w/ FlashAttention |   5.81 |   6.66 |   8.35 |  11.75 | 18.51   | 32.11   |
> | BurstAttention                    |  26.69 |  27.55 |  29.23 |  32.93 | 40.26   | 54.69   |
>
> #### LLama-7b Training Runtime (seconds) on 32xA100 GPUs
>
> | Sequence Length                            | 32768      | 65536      | 98304      | 131072     |
> |:----------------------------------|:-----------|:-----------|:-----------|:-----------|
> | TP(Megatron V3) w/ FlashAttention | 17.14±0.10 | 34.95±0.18 | 55.35±0.46 | 73.99±0.23 |
> | BurstAttention                    | 8.93±0.03  | 18.12±0.13 | 28.91±0.05 | 40.61±0.13 |
> | BurstAttention+ZeRO               | 13.81±0.07 | 23.12±0.09 | 33.56±0.07 | 44.22±0.06 |
>
>
> #### LLama-7b Training Memory Usage (GB) on 32xA100 GPUs
>
> | Sequence Length                            |   32768 |   65536 |   98304 |   131072 |
> |:----------------------------------|--------:|--------:|--------:|---------:|
> | TP(Megatron V3) w/ FlashAttention |    9.89±0.01 |   18.17±0.01 |   26.43±0.01 |    34.70±0.01  |
> | BurstAttention                    |   35.87±0.01 |   46.15±0.01 |   56.24±0.01 |    66.45±0.01 |
> | BurstAttention+ZeRO               |   11.55±0.01 |   21.83±0.01 |   31.92±0.01 |    42.14±0.01 |
>
>
> ### 2.2 The Results of Inferring LLMs
>
> Besides the results of training LLMs, we also show the results of inferring LLMs. In terms of inferring LLMs, BurstAttention still achieves a speed improvement than Megatron V3. Note that, although TP(Megatron V3) is more memory efficient than TP(Megatron V1), the all-reduce operation used by TP(Megatron V1) is better optimized than the reduce-scatter and all-gather operations used by TP(Megatron V3). In the actual inference process, TP(Megatron V1) is even slightly faster than TP(Megatron V3).
>
> #### LLama-7b Inference First Token Latency (seconds) on 8xA100 GPUs
> | Sequence Length                 | 4096      | 8192      | 16384     | 32768     | 65536      | 131072     | 262144     |
> |:-----------------------|:----------|:----------|:----------|:----------|:-----------|:-----------|:-----------|
> | Ring Self-Attention    | 0.42±0.01 | 0.87±0.01 | 2.00±0.01 | 5.13±0.05 | OOM        | OOM        | OOM        |
> | TP(Megatron V1) w/ Flash   | 0.67±0.01 | 1.29±0.01 | 2.58±0.01 | 5.27±0.01 | 11.63±0.02 | 27.54±0.01 | 71.52±0.06 |
> | TP(Megatron V3) w/ Flash   | 0.73±0.02 | 1.36±0.01 | 2.68±0.01 | 5.67±0.01 | 12.25±0.01 | 28.73±0.03 | 75.52±0.05 |
> | BurstAttention w/o LAO | 0.46±0.01 | 0.88±0.01 | 1.79±0.01 | 3.88±0.01 | 10.78±0.01 | OOM        | OOM        |
> | BurstAttention         | 0.44±0.01 | 0.84±0.01 | 1.68±0.01 | 3.27±0.01 | 6.49±0.01  | 16.01±0.01 | 49.32±0.11 |
>
> #### LLama-13b Inference First Token Latency (seconds) on 8xA100 GPUs
> | Sequence Length                 | 4096      | 8192      | 16384     | 32768     | 65536      | 131072     | 262144      |
> |:-----------------------|:----------|:----------|:----------|:----------|:-----------|:-----------|:------------|
> | Ring Self-Attention    | 0.66±0.01 | 1.36±0.01 | 3.08±0.01 | 7.98±0.02 | OOM        | OOM        | OOM         |
> | TP(Megatron V1) w/ Flash   | 1.05±0.01 | 2.01±0.01 | 4.03±0.01 | 8.41±0.01 | 18.56±0.02 | 44.39±0.04 | OOM         |
> | TP(Megatron V3) w/ Flash   | 1.07±0.01 | 2.09±0.01 | 4.20±0.01 | 8.76±0.01 | 19.06±0.06 | 45.46±0.03 | 119.03±0.04 |
> | BurstAttention w/o LAO | 0.72±0.01 | 1.39±0.01 | 2.77±0.05 | 5.99±0.01 | 16.95±0.01 | OOM        | OOM         |
> | BurstAttention         | 0.69±0.01 | 1.40±0.05 | 2.57±0.03 | 5.08±0.02 | 9.92±0.01  | 25.91±0.01 | 78.80±0.07  |
>
> [To be continued]

---

> ### Author Response · Authors · 2023-11-23
> **Response to your questions(Part 4)**
>
> ### 2.3 The Results of Scaling Experiments
>
> Some reviewers suggest us to provide some experimental results of scaling GPU number, sequence length, and batch size. We list the results of scaling experiments in the tables below.
>
> #### LLama-13b Inference First Token Latency (seconds) with Scaling the GPU Number and Sequence Length.
>
> | Setting                   | 1xGPUs SeqLen 16384   | 1xGPUs SeqLen 32768   | 1xGPUs SeqLen 65536   | 2xGPUs SeqLen 16384   | 2xGPUs SeqLen 32768   | 2xGPUs SeqLen 65536   | 4xGPUs SeqLen 16384   | 4xGPUs SeqLen 32768   | 4xGPUs SeqLen 65536   | 8xGPUs SeqLen 16384   | 8xGPUs SeqLen 32768   | 8xGPUs SeqLen 65536   |
> |:-------------------------|:----------------------|:----------------------|:----------------------|:----------------------|:----------------------|:----------------------|:----------------------|:----------------------|:----------------------|:----------------------|:----------------------|:----------------------|
> | TP(Megatron V3) w/ Flash | 3.70±0.01             | 10.90±0.01            | 35.52±0.01            | 4.01±0.01             | 9.85±0.01             | 26.45±0.01            | 3.37±0.01             | 7.52±0.01             | 18.36±0.01            | 4.13±0.01             | 8.88±0.01             | 19.08±0.01            |
> | BurstAttention           | 3.86±0.01             | 11.23±0.01            | 36.35±0.02            | 3.52±0.01             | 8.93±0.01             | 24.83±0.01            | 2.66±0.01             | 6.06±0.01             | 15.77±0.01            | 2.73±0.01             | 5.77±0.01             | 13.05±0.01            |
>
> #### LLama-7b Training Throughput (Token/s) with Scaling the Batch Size
>
> | Setting                 | Batch 1 Seqlen 4096   | Batch 2 Seqlen 4096   | Batch 4 Seqlen 4096   | Batch 8 Seqlen 4096   |
> |:-----------------------|:----------------------|:----------------------|:----------------------|:----------------------|
> | TP(Megatron V3) w/ FlashAttention | 2187.12±25.86         | 2229.40±159.50        | 2203.90±5.73          | 2452.88±4.58          |
> | BurstAttention         | 2521.53±30.34         | 2660.22±70.99         | 2966.01±270.85        | 3525.83±47.52         |
>
>
>
> ### 3. **For the Analysis Section**
>
> Thank you for your insightful suggestion. In our paper, since we mainly focus on the practical performance of BurstAttention, we thus briefly give the communication and memory complexities of different distributed methods and then compare the end-to-end results of these methods. We highly agree that contextualizing our analysis with lower bounds or comparative benchmarks would provide a clearer understanding of BurstAttention's efficiency relative to theoretical optima.
>
> In recent days, we have added more detailed experiments based on reviewer suggestions. In the future, we will further enrich our analysis to include a more detailed examination of the efficiency of BurstAttention, providing insights from a theoretical perspective as well as comparisons with existing benchmarks. These more detailed analyses will be included in our further revisions.
>
>
> We are grateful for the opportunity to address these points and hope that our responses clarify the choices made in our research. We remain open to further suggestions and are committed to continual improvement of our work.

---

> > ### Comment · Area_Chair_fwfb · 2023-12-05
> >
> > Dear Reviewer Gb5f,
> >
> > Please help to read the author's response and make decision. Thanks.
> >
> > Bests, AC

---

> > > ### Comment · Reviewer_Gb5f · 2023-12-05
> > > **Response**
> > >
> > > I updated my review and raised my score in light of the authors' response some days ago; however, as noted, some concerns of mine remain regarding novelty and the depth of the experiments.

---

### Meta-Review · Area_Chair_fwfb · 2023-12-06

**Metareview:**

The paper proposes a parallelism scheme (BurstAttention) for attention to  handle long sequences. BurstAttention divides long sequences into partitions across distributed clusters and employs global and local attention optimization strategies to optimize memory access and communication operations. Experimental results show the effectiveness of  BurstAttention on reducing communication overheads.

The main strength of this work is that it is very simple and efficient. The main issues of three  these reviewers include three aspect. 1)  At least two reviewers think that the novelty of this work is not high, since it combines existing RingAttention, online softmax and FlashAttention together. 2) Experiments are insufficient, e.g. no experiments on large models, and no evaluations on various types of model architectures. Moreover importantly, its "attention-masking"  implementation of Megatron V3+FlashAttention is not fair which is agreed with the authors. Although the authors add many new experiments to test Megatron V3+FlashAttention, it needs to change the manuscript a lot, which means the manuscript is not ready.  Considering most reviewers have low intentions to accept this work, we cannot accept it.

**Justification For Why Not Higher Score:**

The main issues of three  these reviewers include three aspect.

1)  At least two reviewers think that the novelty of this work is not high, since it combines existing RingAttention, online softmax and FlashAttention together.

2) Experiments are insufficient, e.g. no experiments on large models, and no evaluations on various types of model architectures. Moreover importantly, its "attention-masking"  implementation of Megatron V3+FlashAttention is not fair which is agreed with the authors. Although the authors add many new experiments to test Megatron V3+FlashAttention, it needs to change the manuscript a lot, which means the manuscript is not ready.

**Justification For Why Not Lower Score:**

N/A

---

### Decision · Program_Chairs · 2024-01-16

Reject